# External Validation of a Mammography-Derived AI-Based Risk Model in a U.S. Breast Cancer Screening Cohort of White and Black Women

**DOI:** 10.3390/cancers14194803

**Published:** 2022-09-30

**Authors:** Aimilia Gastounioti, Mikael Eriksson, Eric A. Cohen, Walter Mankowski, Lauren Pantalone, Sarah Ehsan, Anne Marie McCarthy, Despina Kontos, Per Hall, Emily F. Conant

**Affiliations:** 1Center for Biomedical Image Computing and Analytics (CBICA), University of Pennsylvania, Philadelphia, PA 19104, USA; 2Department of Radiology, Perelman School of Medicine, University of Pennsylvania, Philadelphia, PA 19104, USA; 3Mallinckrodt Institute of Radiology, Washington University School of Medicine, St. Louis, MO 63110, USA; 4Department of Medical Epidemiology and Biostatistics, Karolinska Institutet, 171 77 Stockholm, Sweden; 5Department of Biostatistics, Epidemiology & Informatics, Perelman School of Medicine, University of Pennsylvania, Philadelphia, PA 19104, USA; 6Department of Oncology, Södersjukhuset, 118 83 Stockholm, Sweden; 7Department of Radiology, Hospital of the University of Pennsylvania, Philadelphia, PA 19104, USA

**Keywords:** breast cancer risk, artificial intelligence, digital mammography, screening, supplemental screening, breast density, racial disparities

## Abstract

**Simple Summary:**

The aim of this study was to perform an external validation in a U.S. screening cohort of a mammography-derived AI risk model that was originally developed in a European study setting. The AI risk model was designed to predict short-term breast cancer risk toward identifying women who could benefit from supplemental screening and/or a shorter screening interval due to their high risk of breast cancer. The AI risk model showed a discriminatory performance of AUC 0.68, comparable to previously reported European validation results (AUC = 0.73). The discriminatory performance of the AI risk model was non-significantly different by race (AUC for White women = 0.67 and for Black women = 0.70), *p* = 0.20. In relation to a clinically used lifestyle–family-based risk model, the AI risk model showed a significantly higher discriminatory performance (AUCs 0.68 vs. 0.55, *p* < 0.01).

**Abstract:**

Despite the demonstrated potential of artificial intelligence (AI) in breast cancer risk assessment for personalizing screening recommendations, further validation is required regarding AI model bias and generalizability. We performed external validation on a U.S. screening cohort of a mammography-derived AI breast cancer risk model originally developed for European screening cohorts. We retrospectively identified 176 breast cancers with exams 3 months to 2 years prior to cancer diagnosis and a random sample of 4963 controls from women with at least one-year negative follow-up. A risk score for each woman was calculated via the AI risk model. Age-adjusted areas under the ROC curves (AUCs) were estimated for the entire cohort and separately for White and Black women. The Gail 5-year risk model was also evaluated for comparison. The overall AUC was 0.68 (95% CIs 0.64–0.72) for all women, 0.67 (0.61–0.72) for White women, and 0.70 (0.65–0.76) for Black women. The AI risk model significantly outperformed the Gail risk model for all women *p* < 0.01 and for Black women *p* < 0.01, but not for White women *p* = 0.38. The performance of the mammography-derived AI risk model was comparable to previously reported European validation results; non-significantly different when comparing White and Black women; and overall, significantly higher than that of the Gail model.

## 1. Introduction

Breast cancer is the most commonly diagnosed cancer among women and is linked with considerable years of life lost (14.9 million DALYs), leading to increased cancer-related morbidity and mortality worldwide. Although mammographic screening reduces breast cancer mortality, a proportion of breast cancers are not detected at mammographic screening and are diagnosed later at a more advanced stage. Therefore, increasing attention is being given to new, personalized approaches to breast cancer screening in which both screening interval and modalities are tailored to an individual woman’s risk based on both clinical and imaging data [1]. 

It is well known that women with the highest levels of mammographic breast density have 3–5 times the risk of developing breast cancer compared to women with lowest breast density [2,3]. In addition, increased mammographic density is associated with decreased mammographic sensitivity due to “masking” of cancers by dense breast tissue [4]. Women with increased mammographic density are often referred for supplemental screening with either ultrasound or magnetic resonance imaging (MRI) [5]. The most frequently used breast cancer risk models, the Gail and Tyrer–Cuzick models, require demographic or other information that are not always readily available, and these risk models have demonstrated only low to moderate prediction performance [6,7]. The construction of risk models using computational imaging data, beyond just breast density, extracted from full-field digital mammography (FFDM) images has the potential to be a viable alternative to traditional models with improved prediction performance [8,9,10,11,12].

In the last 7 years, deep learning, the cornerstone of today’s artificial intelligence (AI) revolution in computational medical imaging, has pervaded mammographic screening as one of the most promising computerized imaging tools [9,10,13,14]. However, the use of AI in clinical practice raises critical questions with regard to algorithm bias. Recent research has shown that AI algorithms developed using U.S. data have been disproportionately trained on White populations not representative of the entire nation [9,15], raising concern that these algorithms may generalize poorly thereby, highlighting the importance of validation across racially diverse screening populations [16].

Acknowledging the critical need to adequately evaluate AI-based breast cancer risk models on heterogeneous screening populations, this study aimed to perform external validation of a commercially available FFDM-derived AI risk model (ProFound AI^®^ Risk 1.0, iCAD Inc., Nashua, NH, USA), originally developed and validated using data from Swedish screening cohorts [17]. To this end, we evaluated the performance of the AI risk model [17] in a cohort of White and Black women undergoing mammographic screening in the United States (U.S.), while also comparing the AI risk model to a clinically established risk model. In the main analysis, we assessed model performance in the overall population and by racial subgroups, and in a sub-analysis, we assessed the model in study subgroups of breast density and tumor subtypes.

## 2. Materials and Methods

### 2.1. Study Design and Data Acquisition

In this institutional review board-approved, Health Insurance Portability and Protection Act (HIPAA)-compliant study under a waiver of consent, we retrospectively analyzed a case–control sample nested within the breast screening practice at the Hospital of the University of Pennsylvania (Figure 1). For the purposes of this study, relying on FFDM images, we focused on all women presenting for annual FFDM screening (Selenia or Selenia Dimensions; Hologic) at our institution between 9/1/2010 and 1/6/2015. Eligible breast cancer cases were derived from all women with a breast cancer diagnosis (with associated biopsy-confirmed tumor pathology via site, and NJ, PA, and DE tri-state registry) after negative or benign mammographic screening 3 months to 2 years prior to cancer diagnosis (*n* = 182). We also identified a random sample of controls (*n* = 4997), defined as women who had mammographic screening studies resulting in negative or benign exams, with at least one-year of screening follow-up without a cancer diagnosis. 

For each cancer case and control, all views of the FFDM ‘For presentation’ imaging data were ascertained. Moreover, all available clinical risk factor data, such as age, race/ethnicity, body mass index (BMI), menopausal status, parity, BI-RADS density category (4th Edition), family history of breast cancer, number of previous breast biopsies, and history of atypical hyperplasia, as well as Gail 5-year risk scores were collected from medical records. For cancer cases, tumor characteristics, such as tumor size, nodal status, metastasis, stage, grade, ER status, and HER2 status, were also ascertained when available.

### 2.2. Short-Term Risk Assessment 

ProFound AI Risk is a short-term risk prediction software that identifies women that have a high likelihood of being diagnosed with breast cancer within 2 years [17]. The KARMA cohort, consisting of ~70,000 women followed for an average of 8 years, was used in developing and validating ProFound AI Risk [17]. The primary model of ProFound AI Risk includes age and imaging features extracted from FFDM images, such as quantified breast density [18] and the presence of masses, microcalcifications and asymmetries of these features between left and right breasts. For this study, we used the 2-year risk scores, and all mammographic features considered the in 2-year risk score calculations (breast percent density, masses score, microcalcifications score, and asymmetry scores) obtained with the primary model of ProFound AI Risk, henceforth “AI risk model”. FFDM exams with indications of failed processing by the AI risk model and exams with negative AI risk scores (patient age > 84 years) were excluded (Figure 1).

### 2.3. Statistical Analysis

Baseline characteristics of the study participants were summarized by standard descriptive summaries including means, standard deviations, and study group differences. Absolute 2-year AI risks were estimated and reported as means in four National Institute of Health and Care Excellence (NICE) guidelines risk categories [19]. Risk stratification performance was based on the NICE defined general, moderate, and high-risk categories while adding a fourth low-risk category and reported as the ratio between low, moderate, and high compared to the NICE general risk category. The high- and low-risk categories were also compared. Case–control discriminatory performance was assessed via age-adjusted area under the ROC curve (AUC) for the entire population [20] as well as for the largest racial subgroups (White and Black). In a sub-analysis, we assessed the model’s discriminatory performance in the study subgroups of the BI-RADS 4th ed. density categories (1–4) and tumor subtypes. Confidence intervals were estimated using bootstrapping. Permutation tests were performed to test for differences between AUCs in the study subgroups. The AI risk model was compared to the Gail 5-year risk model [21] on the subset of the dataset for which Gail risk factors were available, reporting on absolute risks, risk stratification and case–control discriminatory performances. The Gail 5-year risks were categorized into the corresponding four NICE guideline risk categories, which made it possible to compare the proportions of the risk groups using the Gail risk 5-year model with the proportions of the risk groups using the AI 2-year risk model. Moreover, since the Gail risk model has been calibrated mostly for invasive breast cancer, comparisons were also performed separately for invasive breast cancer cases only. A two-sided *p* < 0.05 was indicative of a statistically significant difference.

We also performed an exploratory analysis focusing on potential variation in the discriminatory performance of the AI risk model over images obtained during the same mammographic exam of a women, however acquired at different acquisition time points. The rationale is that, typically, a mammographic exam consists of four FFDM images, with two views of each breast: a cranio-caudal (CC) and a mediolateral oblique (MLO) view of both the right and left breasts. However, multiple images in the same view or projection may be needed, mainly for two reasons: additional imaging may be necessary to adequately image large breasts; second, images may be repeated due to technical issues such as motion or image artifacts. Since the AI risk model uses one image per FFDM view [17], previously, by default, the image used was the image that was acquired first in each view. In this exploratory analysis, the AI risk model was evaluated in two settings: (1) using the first image acquired for each routine FFDM view of each breast (default) and (2) using the images acquired last for each routine FFDM view of the screening exam. 

## 3. Results

### 3.1. Study Dataset Characteristics

The study dataset was composed of 176 women diagnosed with breast cancer (mean age, 59 years; standard deviation, 11 years) and 4963 controls (mean age, 56 years; standard deviation, 10 years). There were statistically significant differences in age at screening (*p* = 0.002), family history of breast cancer (*p* < 0.001), number of prior biopsies (*p* < 0.001), and BI-RADS density categories (*p* < 0.001) between breast cancer cases and controls, but there were no statistically significant differences in BMI, race, menopausal status, parity, or atypical hyperplasia (Table 1). The study dataset consisted primarily of White and Black women, 42% and 51%, respectively (Table 1 and Appendix A). Baseline characteristics by racial groups are provided in Appendix A. The study outcome of cancer detection and tumor characteristics for all cancer cases and for racial subgroups, are available in Appendix A.

### 3.2. External Validation of the AI Risk Model

The generated AI risk scores as well as all key mammographic features considered in AI risk score calculations were found to be higher in breast cancer cases compared to controls (AI risk score: *p* < 0.001, breast percent density: *p* < 0.001, masses malignancy and asymmetry scores: *p* = 0.001, microcalcifications malignancy and asymmetry scores: *p* < 0.001) (Table 2 and Appendix A). Overall, the AI risk model demonstrated an AUC for all women of 0.68 95% CIs [0.64, 0.72] (Table 3), comparable to its performance in the development Swedish cohort (AUC = 0.73 [0.71, 0.74]) [17]. The performance was non-significantly different by race (AUC for White = 0.67 [0.61, 0.72] (cases = 85, controls = 2069) and for Black = 0.70 [0.65, 0.76] (cases = 81, controls = 2521)), *p* = 0.20. In the subgroup analyses, we noted that discriminatory performance differences appear to be driven primarily by small invasive tumors and in situ cancers; however, our analysis was underpowered to fully investigate such differences by cancer subtype (Table 3). Nonsignificant variations in AUC were also observed by breast density (AUC for BI-RADS 4th edition categories 1 + 2 = 0.67 [0.62, 0.72] (cases = 97, controls = 3439) and for 3 + 4 = 0.69 [0.62, 0.74] (cases = 79, controls = 1524)) (Table 3).

Figure 2 shows the distribution of 2-year absolute AI risk and risk categorization using the NICE guidelines in breast cancer cases and control participants in the entire cohort as well as for the two largest racial subgroups. Approximately 21% of the women fell into the highest risk category (women with risk >1.6%) and 1.1% of women fell into in the lowest risk category (risk below 0.15%). The average absolute risk of breast cancer within 2 years in the low-risk group was 0.10%. For the high-risk group, the corresponding value was 3.57%, corresponding to approximately one woman per 28 diagnosed with breast cancer within 2 years. The relative risks of the high- and low-risk groups compared with the reference general-risk group were 8.8 and 0.24, respectively, corresponding to a 37-fold relative risk between high-risk and low-risk women. The corresponding numbers for White and Black women were 36-fold and 34-fold, respectively.

### 3.3. Comparisons with the Gail Risk Model

The Gail risk scores were also higher in breast cancer cases compared to controls (*p* < 0.001) (Table 2). On the subset of the dataset for which Gail risk factors were available (cases = 166, controls = 4894), the AI risk model significantly outperformed the Gail risk model (AUC = 0.68 vs. AUC = 0.55, *p* < 0.01) for any breast cancer type and for invasive breast cancer (AUC = 0.70 vs. AUC = 0.55, *p* < 0.01) (Table 4). Moreover, 2.3% were identified as high-risk based on Gail, and high-risk women were at 18-fold higher risk compared with women at low risk (Figure 3). The corresponding number for AI risk was 20.9% and 36-fold, respectively. Performance differences between the two risk models were significant in Black women (AUC = 0.71 vs. AUC = 0.48, *p* < 0.01; cases = 80, controls = 2487) but not in White (AUC = 0.66 vs. AUC = 0.61, *p* = 0.38; cases = 78, controls = 2037) women (Table 4).

### 3.4. Exploratory Analysis on Potential Effects of FFDM Views on AI Risk

In this study dataset, 1772 of 5139 women (66 cases and 1706 controls) had more than four FFDM images per mammographic exam (Appendix A). The histogram of the number of FFDM images per exam by race suggests that multiple FFDM views were more frequently acquired for Black women, mainly due to larger breast size and higher BMI (Appendix A). The evaluation of the AI risk model on this subset of the study dataset showed that its discriminatory performance is robust over images on the same woman at different acquisition time points during the same mammographic exam. Using the images acquired first (firsst acquisition timepoint) and last (last acquisition timepoint) for each FFDM view of the mammographic exam, the AI risk model demonstrated AUCs of 0.69 95% CIs [0.62, 0.75] and 0.71 95% CIs [0.64, 0.77], respectively (Appendix A). Moreover, when racial subgroups were investigated, we observed similar discriminatory performances between the two settings of the AI risk model, as well as consistent racial differences in AUCs (Appendix A).

## 4. Discussion

We performed an external validation of an AI 2-year risk model for full-field digital mammography (FFDM) in a U.S. screening cohort with White and Black women, including 176 incident breast cancers and approximately five thousand controls. The risk stratification performance of the AI risk model was comparable to previously reported European validation results; non-significantly different when comparing White and Black women in the U.S. study; and overall, significantly higher compared to that of the established Gail risk model. The AI risk model also showed robust discriminatory performance over images of the same mammographic exam acquired on the same woman at different time points.

The use of mammography-derived AI algorithms provides new possibilities for performing breast cancer risk assessment in an imaging clinic [9,10]. The existing mammography screening infrastructure is currently used for cancer detection, but it also entails additional rich information for use in risk assessment of future breast cancers. Traditional risk models that are currently available require lifestyle risk factors, family history of breast cancer, and potentially germline variants to perform risk assessment [21,22]. The performance of a risk assessment model is dependent on the completeness and accuracy of the risk information that is provided to the model. Any missing data or recall bias of self-reported items results in inconsistent risk assessments. In contrast, a mammography-derived AI risk model using a single source of image information could provide more consistent, widely available risk assessments in clinical practice and could potentially also reduce the need and the cost for acquiring risk information.

Traditional risk models such as Gail and Tyrer-Cuzick predict 5-year, 10-year, or lifetime risk and are commonly reported to have low to moderate discriminatory performance [23]. In comparison, several mammography-derived risk models predict 1- to 5-year risk and have reported on moderate to fair discriminatory performance [9,17,24,25]. In our study, we found a higher overall discriminatory performance using the mammography-derived AI risk model compared to the Gail risk model, and we found a low dependency on racial subgroups for the mammography-derived model but a more pronounced racial dependency using the Gail risk model. In a previous study, we did not find a strong racial dependency for Gail 5-year risk in White and Black women [26]. The differential racial dependency using the Gail risk model in our study could possibly be related to the 1-year follow-up in our current study versus the 5-year follow-up in the previous study. Short-term risk assessment of breast cancer is of particular importance for predicting women who are at increased risk of interval cancers and later stage cancers [27]. The identification of cancers at an earlier point in time has the potential to improve survival in breast cancer [28]. 

When comparing the AI risk model performance in our U.S. population to the previous Swedish population [17], we noted similar discriminatory performances for invasive breast cancers, but a point estimate reduction for in situ tumors in the U.S. population. We also observed a tendency of lower model discriminatory performances with smaller tumor sizes in the U.S. population in White women but not in Black women. There could be several reasons for the indicated differences between the two studies. The AI risk model was trained in Sweden using mammograms from the GE, Philips, Sectra, and Siemens vendors, while our U.S. validation set was performed using mammograms from Hologic. In the Swedish study, 12% of the breast cancers were in situ tumors. In the current U.S. study, 27% of the breast cancers were in situ tumors. The tumor sizes at the time of detection were also smaller in White women in the U.S. compared to what is reported in Swedish studies. Annual screening and single-reading are performed in the U.S., while biennial screening and double-reading is performed Sweden [29,30]. The recall rate is ~10% in the U.S. compared to ~3% in Sweden. Supplemental screening with either ultrasound or breast MRI is often performed more frequently in the U.S., while supplemental screening is not frequently performed in Swedish screening. The differences in screening settings may indicate that the mammography-derived AI risk model could benefit from improved performance after the adaption to the U.S. screening routines and population.

Our study was limited since the external validation of the AI risk model was performed with data from only one site in the U.S. Moreover, the comparison to clinically established risk models was restricted to the Gail risk model due to the lack of available risk factors required for using the Tyrer–Cuzick risk model. In our case–control study design, we estimated the 2-year risk using the AI risk model and the 5-year risk using the Gail model in a group of women who were followed for one year on average. Therefore, we could not assess the calibration of the two risk models. The study sample size was a limiting factor for observing potential significant differences in study subgroups including tumor characteristics and ethnicity. In addition, the women in our study were examined on screening machines that acquired both FFDM and digital breast tomosynthesis (DBT) images. The inclusion of DBT imaging may have detected more cancers at screening and, therefore, could have affected the reported discriminatory performance, which was based on FFDM images alone.

## 5. Conclusions

Our preliminary external validation results suggest a promising performance of the AI risk model in a U.S. screening cohort of White and Black women and a clinically meaningful improvement over the established Gail risk model for identifying women at high risk of breast cancer. A mammography-derived risk assessment approach could provide an efficient way to identify women who may benefit from additional clinical follow-up or supplemental screening following a non-actionable screening exam. Future work will include further validation of the AI risk model, as well as its latest extension for DBT [31], at multiple screening sites with ethnically diverse screening populations.

## Figures and Tables

**Figure 1 cancers-14-04803-f001:**
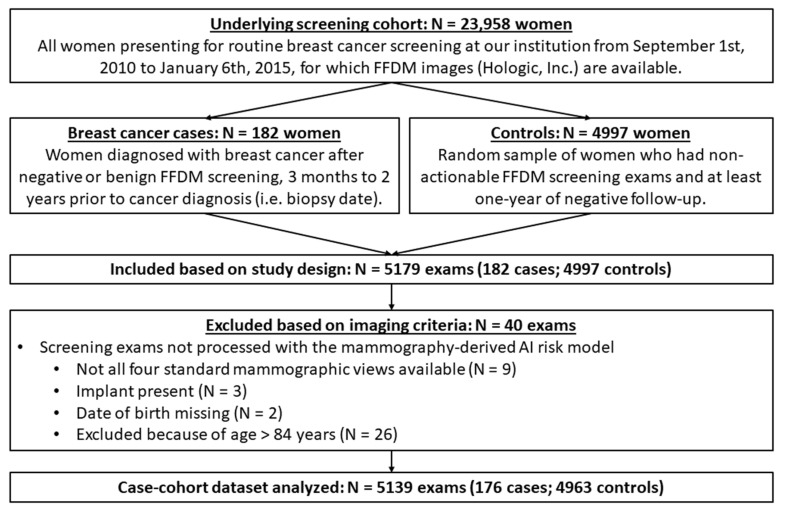
Flowchart showing criteria for case–cohort sample selection. FFDM = full-field digital mammography.

**Figure 2 cancers-14-04803-f002:**
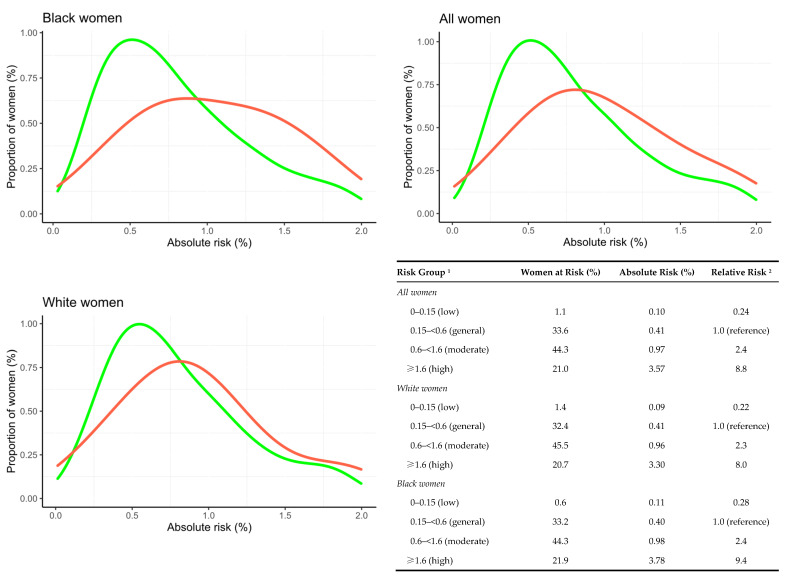
Frequency distribution of AI absolute 2-year risk scores for developing breast cancer in cases (red) and controls (green). Distributions presented for the entire dataset and in racial subgroups. ^1^ Cut-offs for general, moderate, and high-risk groups are based on the NICE guidelines for 10-year risk in age group 40–50 (<3%, 3–8%, >8%) divided by 5. We added a fourth low-risk group with the absolute risk cut-off 0.15. ^2^ The relative risk was calculated as ratios of average risks in each absolute risk category. High-risk women in the full cohort had a 37-fold higher risk compared with women at low risk. The corresponding numbers for White and Black women were 36-fold and 34-fold. NICE: National Institute of Health and Care Excellence guidelines.

**Figure 3 cancers-14-04803-f003:**
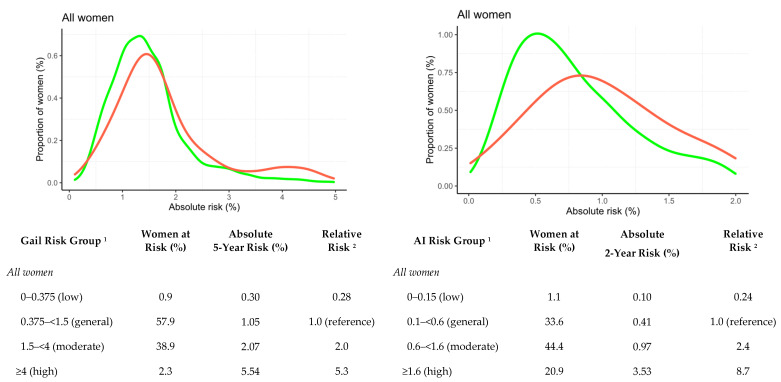
Frequency distribution of Gail 5-year (left column) and AI 2-year (right column) absolute risk scores for developing breast cancer in cases (red) and controls (green). Distributions presented for a subset of *n* = 166 breast cancer cases and *n* = 4894 controls with available Gail and AI risk scores. ^1^ Cut-offs for general, moderate, and high-risk groups are based on the NICE guidelines for 10-year risk in age group 40–50 (<3%, 3–8%, >8%) adapted to 5-year and 2-year, respectively, by dividing the 10-year risk by 2 and 5. We added a fourth low-risk group with the absolute risk cut-off 0.15 2-year risk (or 0.375 5-year risk). ^2^ The relative risk was calculated as ratios of average risks in each absolute risk category. High-risk women identified using Gail 18-fold higher risk compared with women at low risk. The corresponding numbers for AI Risk was 36-fold. NICE: National Institute of Health and Care Excellence guidelines.

**Table 1 cancers-14-04803-t001:** Baseline characteristics of study dataset by case–control status.

Characteristic	Controls, *n* = 4963 ^1^	Cases, *n* = 176 ^1^	*p*-Value ^2^
Age at screening	56.49 (10.32)	59.20 (11.06)	0.002
BMI at screening	29.42 (7.47)	29.37 (6.85)	0.92
Missing BMI	165	10	
Age > 50 (postmenopausal)	3462/4963 (70%)	132/176 (75%)	0.14
Race			0.21
White	2069/4917 (42%)	85/175 (49%)	
Black	2521/4917 (51%)	81/175 (46%)	
Other	327/4917 (6.7%)	9/175 (5.1%)	
Missing	46	1	
Age at first child			0.70
Nulliparous	1102/4302 (26%)	33/142 (23%)	
<20	830/4302 (19%)	24/142 (17%)	
20–24	859/4302 (20%)	29/142 (20%)	
25–29	811/4302 (19%)	27/142 (19%)	
≥30	700/4302 (16%)	29/142 (20%)	
Missing	661	34	
Family history of breast cancer			<0.001
No family history	3985/4899 (81%)	115/167 (69%)	
One 1st degree relative	832/4899 (17%)	39/167 (23%)	
≥2 1st degree relatives	82/4899 (1.7%)	13/167 (7.8%)	
Missing	64	9	
Number of prior biopsies			<0.001
0	438/1227 (36%)	4/46 (8.7%)	
1	543/1227 (44%)	24/46 (52%)	
2 or more	246/1227 (20%)	18/46 (39%)	
Missing	3736	130	
Atypical hyperplasia	31/350 (8.9%)	3/17 (18%)	0.20
Missing	4613	159	
BI-RADS density			<0.001
1	623/4963 (13%)	13/176 (7.4%)	
2	2816/4963 (57%)	84/176 (48%)	
3	1424/4963 (29%)	77/176 (44%)	
4	100/4963 (2.0%)	2/176 (1.1%)	

^1^ Mean (SD); n/N (%). ^2^ For age and BMI, the Welch Two Sample t-test was used; for race, age at first child, family history of breast cancer, number of prior biopsies, and BI-RADS density, the Pearson’s Chi-squared test was used; for postmenopausal status and atypical hyperplasia, the Fisher’s exact test was used.

**Table 2 cancers-14-04803-t002:** AI and Gail risk scores in study dataset: Distributions by case–control status.

Characteristic	Controls, *n* = 4963 ^1^	Cases, *n* = 176 ^1^	*p*-Value ^2^
Breast percent density ^3^	25.88 (20.42)	31.59 (22.14)	<0.001
Calcs malignancy	0.13 (0.16)	0.22 (0.22)	<0.001
Masses malignancy	0.18 (0.19)	0.24 (0.24)	0.001
Calcs asymmetry	0.03 (0.05)	0.07 (0.09)	<0.001
Masses asymmetry	0.05 (0.06)	0.08 (0.08)	<0.001
AI absolute 2-year risk (%)	0.79 (0.49, 1.35)	1.39 (0.79, 2.96)	<0.001
Gail absolute 5-year risk (%) ^4^	1.38 (1.01, 1.76)	1.57 (1.24, 2.21)	<0.001

^1^ Mean (SD); n/N (%); Median (Q1, Q3). ^2^ The Welch Two Sample *t*-test was used for breast percent density, calcs and masses malignancies, and calcs and masses asymmetries; the Wilcoxon rank sum test was used for the AI and Gail absolute risk scores. ^3^ For one breast in a control exam, the percent density was not obtained, and the unilateral density result was used for the risk analysis. ^4^ Gail risk was available on 166 cases and 4894 controls.

**Table 3 cancers-14-04803-t003:** Discriminatory performance (AUC) in the full cohort and in subgroups of women by mammographic density and tumor characteristics, stratified by White and Black women.

Study Participant Characteristic Subgroup	All Women ^1^			White Women			Black Women			*p*-Value ^2^
	*n*	AUC	95% CI	*n*	AUC	95% CI	*n*	AUC	95% CI	
Full cohort	176/4963	0.68	0.64–0.72	85/2069	0.67	0.61–0.72	81/2521	0.70	0.65–0.76	0.20
BI-RADS density										
1 + 2	97/3439	0.67	0.62–0.72	43/1276	0.66	0.58–0.73	48/1975	0.69	0.62–0.76	0.17
3 + 4	79/1524	0.69	0.62–0.74	42/793	0.68	0.60–0.76	33/546	0.71	0.61–0.80	0.69
*p*-value ^3^		0.82			0.85			0.63		
Tumor invasiveness										
Invasive	128/4963	0.70	0.65–0.74	59/2069	0.68	0.60–0.75	62/2521	0.72	0.66–0.78	0.22
In situ	48/4963	0.63	0.55–0.70	26/2069	0.64	0.54–0.74	19/2521	0.65	0.52–0.77	0.74
*p*-value ^3^		0.18			0.64			0.38		
Tumor size (invasive tumors only)										
≤10 mm	68/4963	0.66	0.60–0.72	37/2069	0.63	0.53–0.72	25/2521	0.71	0.62–0.80	0.08
>10–20 mm	38/4963	0.73	0.64–0.81	16/2069	0.73	0.60–0.84	21/2521	0.71	0.59–0.82	0.68
>20 mm	22/4963	0.76	0.67–0.84	6/2069	0.79	0.62–0.91	16/2521	0.74	0.63–0.84	0.95
*p*-value ^3^		0.26			0.11			0.71		
In situ grade										
Low–intermediate	35/4963	0.63	0.54–0.71	18/2069	0.65	0.53–0.77	15/2521	0.60	0.46–0.74	0.55
High	13/4963	0.64	0.48–0.78	8/2069	0.63	0.46–0.77	4/2521	0.83	0.66–0.95	0.12
*p*-value ^3^		0.63			0.37			0.14		

^1^ All women in the cohort also includes non-White and non-Black women, and women with missing information on race. AUCs adjusted for age at baseline. Confidence intervals estimated using bootstrapping. Permutation test tested for difference between AUCs in White and Black women (*p*-value ^2^) and between AUCs in study participant characteristic subgroups (*p*-value ^3^).

**Table 4 cancers-14-04803-t004:** Discriminatory performance (AUC) in women with available Gail risk factors, in the full cohort and in racial subgroups, for any breast cancer subtype and for invasive breast cancer.

Risk Model in Cancer Subgroups	All Women (166/4894) ^1^		White Women (78/2037)		Black Women (80/2487)		*p*-Value ^2^
	AUC	95% CI	AUC	95% CI	AUC	95% CI	
*All cancers*							
Gail 5-year risk	0.55	0.50–0.60	0.61	0.54–0.68	0.48	0.41–0.54	0.12
AI 2-year risk	0.68	0.64–0.72	0.66	0.60–0.72	0.71	0.65–0.76	0.54
*p*-value ^3^	<0.01		0.38		<0.01		
*Invasive cancers*							
Gail 5-year risk	0.55	0.50–0.61	0.61	0.53–0.69	0.47	0.39–0.54	0.12
AI 2-year risk	0.70	0.65–0.75	0.67	0.59–0.74	0.73	0.66–0.79	0.56
*p*-value ^3^	<0.01		0.39		<0.01		

^1^ All women in the cohort also includes non-White and non-Black women, and women with missing information on race. AUCs adjusted for age at baseline. Confidence intervals estimated using bootstrapping. Permutation test tested for difference between AUCs in White and Black women for each model (*p*-value ^2^) and between models (*p*-value ^3^).

## Data Availability

The data generated or analyzed during the study are available from the corresponding author upon reasonable request.

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
