# Peer review of "External Validation of a Mammography-Derived AI-Based Risk Model in a U.S. Breast Cancer Screening Cohort of White and Black Women"

_cancers, 2022, doi:10.3390/cancers14194803_

Round 1

Reviewer 1 Report

Dear Authors:

It is my honor to review the manuscript "External validation of a mammography-derived AI-based risk model in a U.S. breast cancer screening cohort of White and Black women" by Gastounioti et al has demonstrated a promising performance of the AI risk model in a U.S. screening cohort of White and Black women and a clinically meaningful improvement over the established Gail risk model for identifying women at high risk of breast cancer. It is a wonderful manuscript. I just have a few suggestions.

The introduction needs improvement.

If it is possible, please add more basic background information about breast cancer, such as incidence, mortality and so on to highlight the importance of your research. Some previous literature has reported it(1. Advances in the Prevention and Treatment of Obesity-Driven Effects in Breast Cancers. Front Oncol. 2022 Jun 22;12:820968. doi: 10.3389/fonc.2022.820968. PMID: 35814391; PMCID: PMC9258420.

2. Mitochondrial mutations and mitoepigenetics: Focus on regulation of oxidative stress-induced responses in breast cancers. Semin Cancer Biol. 2022 Aug;83:556-569. doi: 10.1016/j.semcancer.2020.09.012. Epub 2020 Oct 6. Erratum in: Semin Cancer Biol. 2022 Jul 16;: PMID: 33035656.)

Best,

Author Response

  • It is my honor to review the manuscript "External validation of a mammography-derived AI-based risk model in a U.S. breast cancer screening cohort of White and Black women" by Gastounioti et al has demonstrated a promising performance of the AI risk model in a U.S. screening cohort of White and Black women and a clinically meaningful improvement over the established Gail risk model for identifying women at high risk of breast cancer. It is a wonderful manuscript. I just have a few suggestions.

Thank you very much for your kind comments about our manuscript. Replies to the suggestions made by the Reviewer are provided below.

  • The introduction needs improvement. If it is possible, please add more basic background information about breast cancer, such as incidence, mortality and so on to highlight the importance of your research. Some previous literature has reported it (1. Advances in the Prevention and Treatment of Obesity-Driven Effects in Breast Cancers. Front Oncol. 2022 Jun 22;12:820968. doi: 10.3389/fonc.2022.820968. PMID: 35814391; PMCID: PMC9258420. 2. Mitochondrial mutations and mitoepigenetics: Focus on regulation of oxidative stress-induced responses in breast cancers. Semin Cancer Biol. 2022 Aug;83:556-569. doi: 10.1016/j.semcancer.2020.09.012. Epub 2020 Oct 6. Erratum in: Semin Cancer Biol. 2022 Jul 16;: PMID: 33035656.)

We have now updated our Introduction section with more background information about breast cancer.

Reviewer 2 Report

A very nice, robust, and well-presented analysis. Well done.

I was under the impression the Gail model predicted invasive cancers and not DCIS. However, I believe the authors classified DCIS as cancer. Or maybe it just wasn’t clear. Here are a couple of publications that support that statement:

https://breast-cancer-research.biomedcentral.com/articles/10.1186/s13058-015-0653-5: “The Gail model was originally developed using a case–control study of women attending screening in the United States [4] with invasive and ductal carcinoma in situ (DCIS) cases, but the absolute rates are calibrated to invasive cancer.”

https://www.ncbi.nlm.nih.gov/pmc/articles/PMC3496137/

-        Supp. Figure 2 indicates 10+ mammographic views, which seem like quite a lot. Where these screening mammograms? Or diagnostic mixed in?

-        The list of authors ends with “and”

-        Where the mammography units used from the same manufacturer as for the European study?

-        I would like to see more information about the 3 months - 2 years. How many where 2 years out compared to 3 months out, etc.?

-        Were the patients aged matched and with similar breast density scores?

-        Did the cancer patients have previous biopsies for benign findings? The authors could expand on what types of benign findings.

-        Why would the patient only go 3 months between screenings if they were thought to be average risk (routine screening)?

-        For the breast density mentioned in line 114-115, are these percent scores or just the A,B,C,D Bi-RADS?

-        The end of section 2.2 mentions the exclusion of older patients. What was the age of the youngest patient?

Author Response

  • A very nice, robust, and well-presented analysis. Well done.

Thank you for your kind feedback about our manuscript.

  • I was under the impression the Gail model predicted invasive cancers and not DCIS. However, I believe the authors classified DCIS as cancer. Or maybe it just wasn’t clear. Here are a couple of publications that support that statement: https://breast-cancer-research.biomedcentral.com/articles/10.1186/s13058-015-0653-5: “The Gail model was originally developed using a case–control study of women attending screening in the United States [4] with invasive and ductal carcinoma in situ (DCIS) cases, but the absolute rates are calibrated to invasive cancer.” https://www.ncbi.nlm.nih.gov/pmc/articles/PMC3496137/

Thank you for the useful comment. Indeed, the Gail model is calibrated to mostly predict invasive breast cancer. To address this point, in the comparisons of our AI risk model with the Gail risk model, we have expanded our analyses to also compare their discriminatory performances specifically for invasive cancer cases (Table 4). Based on our results, the AI risk model significantly outperformed the Gail risk model for all breast cancer subtypes, as well as specifically for invasive breast cancer cases. Sections 2.3 and 3.3 have been updated accordingly.

  • Figure 2 indicates 10+ mammographic views, which seem like quite a lot. Where these screening mammograms? Or diagnostic mixed in?

We would like to clarify that in our study we used screening mammograms only, with no diagnostic mammograms mixed in. Multiple mammographic images may be acquired in screening, mainly for two reasons; additional imaging may be necessary to adequately image large breasts; and second, images may be repeated due to technical issues such as motion or image artifacts. This point is clarified in section 2.3.

  • The list of authors ends with “and”.

This typo has now been corrected.

  • Were the mammography units used from the same manufacturer as for the European study?

The manufacturer of the mammography units in this study was Hologic, while GE, Siemens, Sectra, and Philips mammography units were used in the European study. A related comment has been added in the Discussion section of our manuscript.

  • I would like to see more information about the 3 months - 2 years. How many where 2 years out compared to 3 months out, etc.?

Thank you for commenting on this important point. We have now included more information on the distribution of the time from screening to breast cancer diagnosis, by reporting detailed percentiles in Supplementary Table S2. Out of the 176 breast cancers, 2 breast cancers (the first percentile) were diagnosed between 90 days and 107 days. Half of the breast cancers were diagnosed at least 423 days after the screening mammogram.

  • Were the patients aged matched and with similar breast density scores?

We would like to point out that our study dataset was compiled as a representative sample from the breast cancer screening population at the Hospital of the University of Pennsylvania, reflecting the natural distribution in age and breast density. However, breast density was included as a key mammographic feature considered in AI risk score calculations and we also controlled for age in our analyses, to rule out the possibility of inflated results due to age differences between cases and controls.

  • Did the cancer patients have previous biopsies for benign findings? The authors could expand on what types of benign findings.

Unfortunately, we had no access to that level of detail and therefore reporting on the types of benign findings was not feasible at this moment.

  • Why would the patient only go 3 months between screenings if they were thought to be average risk (routine screening)?

We would like to clarify that the time window of 3 months to 2 years refers to the time from the mammographic screen analyzed to the time of cancer diagnosis, while there was no additional mammographic screen in-between. Also, cancers were either diagnosed due to patient symptoms (lump, discharge, etc.) or, on supplemental screening (please see also Supplementary Table S2).

  • For the breast density mentioned in line 114-115, are these percent scores or just the A,B,C,D Bi-RADS?

We wish to clarify that percent density scores are used in the AI risk score calculations. We have revised this sentence accordingly.  

  • The end of section 2.2 mentions the exclusion of older patients. What was the age of the youngest patient?

The youngest woman in our dataset was 40 years old.

Reviewer 3 Report

Reviewer comments

Regarding the limitation of the traditional risk models and the bias and generalizability of the AI risk models, it is critical to perform further validation to evaluate the AI-based breast cancer risk models on heterogeneous screening populations. In this study, the authors performed external validation of a mammography-derived AI breast cancer risk model and evaluated its performance in a U.S. screening cohort of White and Black women. The external validation results demonstrate a promising performance of the AI risk model and a clinically meaningful improvement over the Gail risk model for identifying women at high risk of breast cancer.

The manuscript is well-written and well-organized. The objective is well-articulated and reached. The results and analysis presented in the manuscript are interesting for this field and Cancers is the appropriate place to submit it. But there are still some points that the authors should consider, as described in the following. Also, some suggestions are provided, in case the authors consider them interesting to carry out.

In the main text, we can see “discriminatory performance” and “discrimination performance”. Do they indicate the same term? Is it possible to keep them consistent?

In Table 1, we can see “missing” shown below BMI at screening in the first block. What does this missing mean? Missing age, BMI, or menopausal status? The same confusion happens in Supplementary Table 1.

In Table 1, it is shown that 46 controls have missing information in race. However, N is 38 in the unknown group of controls in Supplementary Fig. 1. Does the unknown group represent the people with missing race information? In addition, the figure legend in Supplementary Fig. 1 shows missing 8 in controls and missing 0 in cases. But these two missing numbers are 46 and 1 in Table 1. Please keep the data consistent in Table 1 and Supplementary Fig. 1.

In Table 1, Median (IQR) is written at the bottom, but it is not clear to tell if the IQR is represented by a range (Q3-Q1) or two numbers (Q1, Q3) until we see Supplementary Table 2. It would be a plus to clearly introduce that IQR is described using two numbers.

 In Table 1 and Table 2, four statistical tests are listed at the bottom. There are multiple p-values in this table. How to know which p-value is calculated using which statistical test?

In 3.2 line 185-186, AI risk score: p < 0.001, breast density: p < 0.001, masses score: p = 0.001, microcalcifications score: p < 0.001 and asymmetry scores: p < 0.001. These terms of characteristic are different from those in the first column of Table 2 and Supplementary Table 3. Please keep them consistent.

In Supplementary Table 3, please adjust the font size and make sure Mean (SD) or Median (IQR) is kept on the same row (e.g., the row of breast percent density, AI 2-year risk, Gail 5-year risk).

In Fig. 2 and Fig. 3, the authors should consider increasing the font size of x-axis and y-axis labels.

In 3.3 line 226-228, on the subset of the dataset for which Gail risk factors were available (cases=166, controls=4,894), the AI risk model significantly outperformed the Gail risk model (AUC = 0.68 vs AUC = 0.55, p<0.01). Can the authors provide multiple subsets to demonstrate the outperformance of the AI risk model is robust across multiple subsets of dataset?

In line 202, Figure 2 should be changed to Fig. 2 since Fig. 1 and Fig. 3 are used in the main text.

In Supplementary Fig. 3, the right part of this figure seems incomplete.

Supplementary Table 1 and Supplementary Table S1 are used in the main text and the supplementary materials, respectively. Please keep them consistent.

Author Response

  • Regarding the limitation of the traditional risk models and the bias and generalizability of the AI risk models, it is critical to perform further validation to evaluate the AI-based breast cancer risk models on heterogeneous screening populations. In this study, the authors performed external validation of a mammography-derived AI breast cancer risk model and evaluated its performance in a U.S. screening cohort of White and Black women. The external validation results demonstrate a promising performance of the AI risk model and a clinically meaningful improvement over the Gail risk model for identifying women at high risk of breast cancer. The manuscript is well-written and well-organized. The objective is well-articulated and reached. The results and analysis presented in the manuscript are interesting for this field and Cancers is the appropriate place to submit it. But there are still some points that the authors should consider, as described in the following. Also, some suggestions are provided, in case the authors consider them interesting to carry out.

Thank you for your kind comments about our study and the very useful suggestions. Point-by-point replies to the comments raised by the Reviewer are provided below.

  • In the main text, we can see “discriminatory performance” and “discrimination performance”. Do they indicate the same term? Is it possible to keep them consistent?

We apologize for this inconsistency. We are now using the term “discriminatory performance” throughout the manuscript.

  • In Table 1, we can see “missing” shown below BMI at screening in the first block. What does this missing mean? Missing age, BMI, or menopausal status? The same confusion happens in Supplementary Table 1.

We have clarified this point in both Tables.

  • In Table 1, it is shown that 46 controls have missing information in race. However, N is 38 in the unknown group of controls in Supplementary Fig. 1. Does the unknown group represent the people with missing race information? In addition, the figure legend in Supplementary Fig. 1 shows missing 8 in controls and missing 0 in cases. But these two missing numbers are 46 and 1 in Table 1. Please keep the data consistent in Table 1 and Supplementary Fig. 1.

Thank you for bringing this up. We have now updated the erroneous Supplementary Figure S1.

  • In Table 1, Median (IQR) is written at the bottom, but it is not clear to tell if the IQR is represented by a range (Q3-Q1) or two numbers (Q1, Q3) until we see Supplementary Table 2. It would be a plus to clearly introduce that IQR is described using two numbers.

We apologize for the omission, and we have now corrected this in all Tables.

  • In Table 1 and Table 2, four statistical tests are listed at the bottom. There are multiple p-values in this table. How to know which p-value is calculated using which statistical test?

Thank you for pointing this out. We now state which tests were used for which variables in both Tables.

  • In 3.2 line 185-186, AI risk score: p < 0.001, breast density: p < 0.001, masses score: p = 0.001, microcalcifications score: p < 0.001 and asymmetry scores: p < 0.001. These terms of characteristic are different from those in the first column of Table 2 and Supplementary Table 3. Please keep them consistent.

We have now edited the text in section 3.2 to be consistent with all tables as follows.

  • In Supplementary Table 3, please adjust the font size and make sure Mean (SD) or Median (IQR) is kept on the same row (e.g., the row of breast percent density, AI 2-year risk, Gail 5-year risk).

Done.

  • In Fig. 2 and Fig. 3, the authors should consider increasing the font size of x-axis and y-axis labels.

We updated the font size to be more readable. The figure is now also in vector format.

  • In 3.3 line 226-228, on the subset of the dataset for which Gail risk factors were available (cases=166, controls=4,894), the AI risk model significantly outperformed the Gail risk model (AUC = 0.68 vs AUC = 0.55, p<0.01). Can the authors provide multiple subsets to demonstrate the outperformance of the AI risk model is robust across multiple subsets of dataset?

Thank you for this important comment. In response also to comment R2.2. of Reviewer #2, we have now expanded Table 4 to include comparisons in invasive breast cancer.

  • In line 202, Figure 2 should be changed to Fig. 2 since Fig. 1 and Fig. 3 are used in the main text.

Done.

  • In Supplementary Fig. 3, the right part of this figure seems incomplete.

This has been corrected.

  • Supplementary Table 1 and Supplementary Table S1 are used in the main text and the supplementary materials, respectively. Please keep them consistent.

Done.

Round 2

Reviewer 1 Report

Strongly suggest for publication